# Early Anticoagulation in Patients with Acute Ischemic Stroke Due to Atrial Fibrillation: A Systematic Review and Meta-Analysis

**DOI:** 10.3390/jcm11174981

**Published:** 2022-08-25

**Authors:** Lina Palaiodimou, Maria-Ioanna Stefanou, Aristeidis H. Katsanos, Maurizio Paciaroni, Simona Sacco, Gian Marco De Marchis, Ashkan Shoamanesh, Konark Malhotra, Diana Aguiar de Sousa, Vaia Lambadiari, Maria Kantzanou, Sofia Vassilopoulou, Konstantinos Toutouzas, Dimitrios K. Filippou, David J. Seiffge, Georgios Tsivgoulis

**Affiliations:** 1Second Department of Neurology, “Attikon” University Hospital, School of Medicine, National and Kapodistrian University of Athens, 12462 Athens, Greece; 2Division of Neurology, McMaster University and Population Health Research Institute, Hamilton, ON L8L2X2, Canada; 3Stroke Unit and Division of Cardiovascular Medicine, University of Perugia, 06129 Perugia, Italy; 4Neuroscience Section, Department of Applied Clinical Sciences and Biotechnology, University of L’Aquila, 67100 L’Aquila, Italy; 5Neurology and Stroke Center, University Hospital of Basel, University of Basel, 4031 Basel, Switzerland; 6Department of Neurology, Allegheny Health Network, Pittsburgh, PA 15212, USA; 7Stroke Center, Lisbon Central University Hospital, 1649-024 Lisbon, Portugal; 8Institute of Anatomy and CEEM, Faculdade de Medicina, Universidade de Lisboa, 1649-028 Lisbon, Portugal; 9Second Department of Internal Medicine, Research Institute and Diabetes Center, “Attikon” University Hospital, Medical School, National and Kapodistrian University of Athens, 12462 Athens, Greece; 10Department of Hygiene, Epidemiology and Medical Statistics, School of Medicine, National and Kapodistrian University of Athens, 11527 Athens, Greece; 11First Department of Neurology, Eginition Hospital, School of Medicine, National and Kapodistrian University of Athens, 15772 Athens, Greece; 12First Department of Cardiology, Hippokration General Hospital, Medical School, National and Kapodistrian University of Athens, 11527 Athens, Greece; 13Department of Anatomy and Surgical Anatomy, Medical School, National and Kapodistrian University of Athens, 11527 Athens, Greece; 14National Organization for Medicines (EOF), 15562 Athens, Greece; 15Neurology and Stroke Center, Inselspital, University Hospital Bern, 3010 Bern, Switzerland; 16Department of Neurology, University of Tennessee Health Science Center, Memphis, TN 38163, USA

**Keywords:** anticoagulant, vitamin-K antagonists, direct oral anticoagulants, atrial fibrillation, ischemic stroke, secondary prevention, intracerebral haemorrhage

## Abstract

**Introduction:** There is uncertainty regarding the optimal timing for initiation of oral anticoagulation in patients with acute ischemic stroke (AIS) due to atrial fibrillation (AF). **Methods:** We performed a systematic review and meta-analysis of randomized-controlled clinical trials (RCTs) and prospective observational studies to assess the efficacy and safety of early anticoagulation in AF-related AIS (within 1 week versus 2 weeks). A second comparison was performed assessing the efficacy and safety of direct oral anticoagulants (DOACs) versus vitamin-K antagonists (VKAs) in the two early time windows. The outcomes of interest were IS recurrence, all-cause mortality, symptomatic intracerebral haemorrhage (sICH) and any ICH. **Results:** Eight eligible studies (6 observational, 2 RCTs) were identified, including 5616 patients with AF-related AIS who received early anticoagulation. Patients that received anticoagulants within the first week after index stroke had similar rate of recurrent IS, sICH and all-cause mortality compared to patients that received anticoagulation within two weeks (test for subgroup differences *p* = 0.1677; *p* = 0.8941; and *p* = 0.7786, respectively). When DOACs were compared to VKAs, there was a significant decline of IS recurrence in DOAC-treated patients compared to VKAs (RR: 0.65; 95%CI: 0.52–0.82), which was evident in both time windows of treatment initiation. DOACs were also associated with lower likelihood of sICH and all-cause mortality. **Conclusions:** Early initiation of anticoagulation within the first week may have a similar efficacy and safety profile compared to later anticoagulation (within two weeks), while DOACs seem more effective in terms of IS recurrence and survival compared to VKAs.

## 1. Introduction

Atrial fibrillation (AF) is associated with up to five times increased risk of stroke [1], which carries a higher risk of adverse functional outcomes at 3 months, compared to other causes of cardioembolic strokes [2]. Furthermore, AF has been associated with early ischemic stroke recurrence [3] and therefore anticoagulant treatment is indicated as part of secondary stroke prevention [1]. On the other hand, AF-associated ischemic strokes (IS) have also been related with a higher risk of haemorrhagic transformation [4], complicating the treatment decision regarding the optimal timing for initiating anticoagulants after IS. This therapeutic challenge is clearly depicted by the lack of specific recommendations in both the American Heart Association/American Stroke Association [5] and the European Stroke Organization guidelines [6]. The general consensus is to individualize treatment initiation according to each patient’s risk of haemorrhage versus recurrent embolism, with observational evidence suggesting an optimal window of 4–14 days [7,8]. 

Considering that direct oral anticoagulants (DOACs) have been proven to be equally effective with vitamin-K antagonists (VKAs) in preventing IS in patients with AF, yet safer in terms of intracranial haemorrhage (ICH), any major bleeding or death from any cause [6,8,9] means that it is reasonable to ponder whether there are any differences in terms of effectiveness and safety when administered in the early time window after IS (≤14 days). Recently, preliminary findings from randomized-controlled clinical trials (RCTs) on early anticoagulation indicate that DOACs may have similar efficacy to VKAs in IS prevention, but lower risk of haemorrhagic transformation and ICH in AF-related IS [10,11]. 

We sought to systematically collect and synthesize available evidence from RCTs and prospective observational studies in order to assess the efficacy and safety of early anticoagulation when administered within 1 week versus within 2 weeks after index stroke and compare the efficacy and safety of DOACs versus VKAs in this setting.

## 2. Methods

### 2.1. Data Sources, Searches and Study Selection

A systematic literature search was conducted to identify eligible studies reporting on patients with an AF-related AIS or transient ischemic attack (TIA) who were prescribed anticoagulation with either DOACs or VKAs in the early time window post-stroke (≤14 days). The literature search was performed independently by three reviewers (LP, MIS, AHK). We searched MEDLINE, and Scopus, using search strings that included the following terms: “stroke”, “atrial fibrillation”, “direct oral anticoagulants”, “vitamin-K antagonists”, “initiation”, “recurrent stroke”, and “intracerebral hemorrhage”. The complete search algorithms used in MEDLINE and Scopus are provided in the Supplement. Our search spanned from inception of each electronic database to 11 November 2021. No language or other restrictions were applied. We additionally searched reference lists of published articles manually, to ensure the comprehensiveness of the bibliography.

RCTs and observational cohort studies presenting patients that were administered oral anticoagulants early (within 2 weeks) after AF-related IS were considered eligible. The studies had to assess efficacy outcomes (recurrence of IS or arterial embolism) and safety outcomes (ICH, or major bleeding, or death from any cause) in patients receiving DOACs compared to VKAs. Case series, case reports, commentaries, editorials, and narrative reviews were excluded. In cases of overlapping data between studies, the study with the largest dataset was retained. All retrieved studies were independently assessed by three reviewers (LP, MIS, AHK), and any disagreements were resolved after discussion with a fourth tie-breaking evaluator (GT).

### 2.2. Quality Control, Bias Assessment and Data Extraction

Eligible studies were subjected to quality control and bias assessment employing the Cochrane Collaboration tool (RoB 2) [12] for RCTs and the Risk Of Bias In Non-randomized Studies of Interventions (ROBINS-I) tool [13] for cohort studies. Quality control and bias assessment were conducted independently by two reviewers (LP, MIS), and disagreements were settled by consensus after discussion with the corresponding author (GT).

Data extraction was performed in structured reports, including author names, date of publication, study design, country, oral anticoagulant type, patients’ characteristics, and efficacy and safety events. We also contacted the corresponding authors of individual studies to provide unpublished data where appropriate.

### 2.3. Outcomes

An aggregate data meta-analysis was performed with the inclusion of the identified RCTs and cohort studies. There were two main comparisons. The first one was the time window of early anticoagulation, being within 1 week versus 2 weeks after index stroke with the following outcomes of interest: IS recurrence, symptomatic ICH, any ICH, and death from any cause. The second comparison assessed the same efficacy and safety outcomes between DOACs versus VKA. The patients’ characteristics were also evaluated in order to disclose any differences between the two groups, including the: (1) proportion of women; (2) mean age of patients; (3) mean baseline National Institutes of Health Stroke Scale (NIHSS), CHA_2_DS_2_-VASc and HAS-BLED scores, (4) mean baseline infarct volume; and (5) concomitant diseases.

### 2.4. Statistical Analysis

For the aggregate meta-analysis, we calculated for each dichotomous outcome of interest the corresponding pooled proportion with further assessment of the subgroup differences (after the implementation of the variance-stabilizing double arcsine transformation). We also calculated the corresponding risk ratios (RR) and 95% confidence intervals (95%CI) for the comparison of dichotomous outcomes between DOACs and VKAs. For studies reporting continuous outcomes in median values and corresponding interquartile ranges we estimated the sample mean and standard deviation using the quantile estimation method. Continuous outcomes were assessed by mean difference (MD). The random-effects model of meta-analysis (DerSimonian and Laird) was used to calculate the pooled estimates. Study estimates were pooled under the random-effects model [14]. Heterogeneity was assessed with the I^2^ and Cochran Q statistics. For the qualitative interpretation of heterogeneity, I^2^ values > 50% and values > 75% were considered to represent substantial and considerable heterogeneity, respectively. The significance level for the Q statistic was set at 0.1. Publication bias across individual studies was graphically assessed when more than four studies were included in each analysis, using both funnel plot inspection and the Egger’s linear regression test [15] and also the equivalent z test for each pooled estimate where a two-tailed *p* value < 0.05 was considered statistically significant. Finally, to further evaluate the robustness of our results regarding the primary outcomes, sensitivity analyses were conducted by: (i) repeating the analyses after excluding each study (leave-one-out meta-analysis); and (ii) by assessing for potential subgroup differences between the different study designs (RCTs versus observational studies). All statistical analyses were conducted using the Cochrane Collaboration’s Review Manager (RevMan 5.3) Software Package (Copenhagen: The Nordic Cochrane Centre, The Cochrane Collaboration, 2014), the OpenMetaAnalyst [16] and R software version 3.5.0 (package: metafor; R Foundation for Statistical Computing, Vienna, Austria.) [17]. 

## 3. Results

### 3.1. Literature Search and Included Studies

The systematic database search yielded a total of 90 and 91 records from the MEDLINE and SCOPUS databases, respectively (Appendix A). After excluding duplicates and initial screening we retrieved the full text of 19 records that were considered potentially eligible for inclusion. After reading the full-text articles, 11 records were further excluded (Appendix A). Finally, we identified 8 eligible studies for inclusion [10,11,18,19,20,21,22,23], comprising a total of 5616 patients that received oral anticoagulation within 2 weeks following an AF-related IS. Five studies were prospective cohort studies [18,19,20,21,22], one study was an individual patient data analysis and pooled synthesis of seven prospective cohort studies [23], and two studies were RCTs [10,11]. We received unpublished data from the corresponding author of one study [23]. All the studies were included in the single-arm analysis assessing subgroup differences between different time windows (within 1 week versus within 2 weeks), while 3 of them presented direct comparisons between DOACs versus VKAs and were included in the meta-analysis evaluating the comparative efficacy of DOACs and VKAs. The characteristics of the included studies are presented in Table 1.

### 3.2. Quality Control of Included Studies

The risk of bias in included RCTs was assessed by the Cochrane Collaboration tool (RoB 2) [12] and is presented in Appendix A. Overall, the included RCTs were considered of high quality with low risk of bias detected in all individual domains, with the exception of high risk of performance bias, since both studies were open-label with blinded endpoint adjudicators. The risk of bias in the included cohort studies was assessed by the Risk Of Bias In Non-randomized Studies of Interventions (ROBINS-I) tool [13] that is presented in Appendix A. Four [19,20,21,22] out of the six studies included were not controlled, therefore the assessment of confounding bias, bias in classification of intervention and bias due to deviations from intended interventions were not applicable. Yet, serious confounding bias were detected in the two controlled studies [18,23], since there were several significant baseline differences between the patients’ subgroups.

### 3.3. Quantitative Analyses


Single-arm analysis with subgroups of different time windows of anticoagulants initiation (within 1 week versus 2 weeks after the index stroke).


An overview of the results of the single-arm analysis for all primary outcomes is summarized in Table 2. The pooled proportion of recurrent IS among patients that received oral anticoagulation was 5.3% (95%CI: 3.7–7.3%; 7 studies; I^2^ = 75%; *p* for Cochran Q < 0.001; Figure 1). When subgroups were assessed, patients that received anticoagulants within the first week after index stroke had a similar rate of recurrent IS (3.3%; 95%CI: 0.7–7.8%) compared to patients that received anticoagulation within 2 weeks (6.9%; 95%CI: 3.8–10.9%; test for subgroup differences *p* = 0.1677).

sICH was present in 1.3% of the patients (95%CI: 0.8–2.1%; 8 studies; I^2^ = 49%; *p* for Cochran Q = 0.039; Appendix A). Importantly, no subgroup differences between the two different time windows were disclosed (*p* for subgroup differences = 0.8941). When all cases of ICH were considered (including symptomatic and asymptomatic ICH), there was a statistically significant subgroup difference between the group of anticoagulant initiation within 1 week (pooled rate 27.9%; 95%CI: 22.3–33.8%; 3 studies; I^2^ = 19%; *p* for Cochran Q = 0.293; Appendix A) versus within 2 weeks (pooled rate 11.5%; 95%CI: 4.6–21%; 5 studies; I^2^ = 85%; *p* for Cochran Q < 0.001; *p* for subgroup differences = 0.0085; Appendix A). Nevertheless, all-cause mortality was similar between the two subgroups (*p* for subgroup differences = 0.7786; overall pooled rate 4.9%; 95%CI: 2.8–7.5%; 7 studies; I^2^ = 88%; *p* for Cochran Q < 0.001; Appendix A).

Baseline characteristics of the patients included are presented in Appendix A. Forty-eight percent of the patients were women, with a mean age of 73 years, mean NIHSS score of 5, mean CHA_2_DS_2_-VASc score of 4, mean HASBLED score of 3, with a mean baseline lesion volume of 5 mL. With regards to potential comorbidities, 17% of the patients had a history of stroke prior the index event, 77% had hypertension, 33% had dyslipidemia and 27% had diabetes mellitus, while chronic kidney failure was evident in 19% of the patients. Importantly, no subgroup differences were disclosed for any of the baseline characteristics.

Finally, publication bias was evaluated using funnel plots for every primary outcome of the analysis; asymmetry or evidence of small study effects (i.e., publication bias) were uncovered through funnel plot inspection, but were not confirmed by the Egger’s linear regression test for IS recurrence, symptomatic ICH or any ICH. (Appendix A). On the other hand, all-cause mortality presented a significant publication bias (*p* for Egger’s test < 0.001; Appendix A).

Pair-wise analysis comparing DOACs versus VKAs in patients receiving early anticoagulation (within 1 week versus 2 weeks after the index stroke).

An overview of the results of the two-arm analysis for all primary outcomes is summarized in Table 3. There was a significant decline of IS recurrence in patients treated with DOACs compared to VKAs (RR: 0.65; 95%CI: 0.52–0.82; 3 studies; I^2^ = 0%; *p* for Cochran Q = 0.98; Figure 2), which was evident in both time-windows of treatment initiation (within 1 week versus 2 weeks). DOACs were also associated with a lower risk of symptomatic ICH compared to VKAs (RR 0.36; 95%CI: 0.22–0.59; 2 studies; I^2^ = 0%; *p* for Cochran Q = 0.82; Appendix A), which was independent of the timing of initiation. When all cases of ICH were considered, including both symptomatic and asymptomatic ICH, there was no difference between DOACs and VKAs in any time-window of treatment initiation (RR: 1.08; 95%CI: 0.71–1.63; 2 studies; I^2^ = 0%; *p* for Cochran Q = 0.82; Appendix A). Mortality was also lower in the DOAC-treated group (RR: 0.40; 95%CI: 0.25–0.65; 2 studies; I^2^ = 78%; *p* for Cochran Q = 0.01; Appendix A). There was a significant subgroup difference, pointing to lower mortality in patients receiving treatment within the first week after the index event compared to those treated within 2 weeks (*p* for subgroup differences = 0.003).

Baseline characteristics were also compared between the two treatment groups and are presented in the Appendix A. Sex, age, baseline NIHSS, CHA_2_DS_2_-VASc and HAS-BLED scores were well balanced between the two groups (Appendix A). When concomitant diseases were compared, it was found that DOACs were associated with a lower risk of a known history of dyslipidemia; rates of prior history of stroke, hypertension and diabetes mellitus were similar between the two treatment groups (Appendix A).


Sensitivity analyses of the primary outcomes


No statistically significant difference was disclosed after repeating the analyses by excluding each study (leave-one-out) meta-analysis for the primary outcomes of both the single-arm and the pair-wise meta-analysis (Appendix A). Moreover, no subgroup differences were unravelled when analyses were stratified by study-design (RCTs versus observational studies; Appendix A). Sensitivity analyses confirmed the robustness of the results for IS recurrence, sICH, any ICH and mortality.

## 4. Discussion

In the present systematic review and meta-analysis, including data from 5616 AIS patients with AF, we documented a similar IS recurrence risk by initiation of oral anticoagulants within the first week compared to two weeks from index AIS. Anticoagulation within the first week of AIS was not associated with an increased risk of sICH or all-cause mortality. However, there was a higher risk for any ICH (symptomatic or asymptomatic) in patients treated within 1 week versus within 2 weeks after AIS. Notably, cardiovascular risk factors and bleeding risk were similar between patients receiving anticoagulation in the two time-windows (within 1 or 2 weeks) following AIS, while, as reflected by the pooled analysis of patient characteristics (mean NIHSS score and infarct volume of 5 points and 5 mL, respectively), most studies included patients with mild stroke severity and small infarct volumes.

With respect to comparative efficacy of oral anticoagulants, we documented reduced risk of IS recurrence in patients treated with DOACs compared to VKAs, regardless of the timing of treatment initiation (i.e., in both time windows). Additionally, although the risk of any ICH was not related to the type of anticoagulation, DOACs were associated with a significantly lower risk of sICH compared to VKAs that was consistent in both treatment time windows. Overall, treatment with DOACs was associated with significantly improved survival compared to VKAs, while improved survival was associated with early (within 1 week) as opposed to later anticoagulation (within 2 weeks). Finally, within each treatment time-window, DOAC- and VKA-treated patients had similar cardiovascular and haemorrhagic risk profiles.

In line with the recent guidelines of the American Heart Association/American College of Cardiology/Heart Rhythm Society [24], the American Heart Association/American Stroke Association [5] and the European Stroke Organization [6] that provide Class I recommendations (level of evidence A) for the use of DOACs over VKAs for primary and secondary stroke prevention in patients with nonvalvular AF, our findings are supporting DOAC over VKA treatment both with respect to safety and efficacy for IS prevention in the early post-AIS period. Indeed, a pooled analysis of individual patient data derived from eight prospective cohort studies confirmed no safety concerns regarding the use of DOAC in the early time-window (within 5 days post AIS) [25]. With respect to the timing of initiation of anticoagulation, current guidelines provide less explicit recommendations (based on level of evidence B or C) advising initiation of anticoagulant treatment within a broad time-window of 4–14 days, while stratifying patients according to the risk of haemorrhagic transformation mostly based on stroke severity or infarct size. The European Stroke Organization (ESO) guidelines [6], based on expert consensus and indirect observational data, recommend the “4-7-14” day rule for initiating anticoagulation: at day 3 or 4 from the index IS in patients with mild stroke and small infarct size (<1.5 cm), after 7 days in those with moderate stroke, and after 14 days in those with severe stroke and large infarct size (yet, without providing exact definitions for mild, moderate or severe stroke). Another proposed algorithm recommends initiation of anticoagulants as early as within the first two days post AIS, if infarct size is ≤1.5 cm [26]. However, all proposed “day rules” stem from the VKA era and may not be generalized to DOACs.

The findings of the present meta-analysis, including results from two recent RCTs on DOAC versus VKA use for AF-related IS, are aligned with the previous recommendations, indicating a net benefit from early anticoagulation with DOACs within the first week from mild to small-sized AF-related AIS. However, it should be noted that a finer-grained analysis of the timing of anticoagulants initiation within the first week was not possible (due to the limited number of studies, diverse time-windows for commencing anticoagulation, and unavailability of individual patient data). On the other hand, it may be argued that there are accruing data suggesting that initiation of anticoagulation may be considered even earlier (i.e., within ≤3 days of IS onset) [9,27] in AF-patients at low risk of haemorrhage [6]. 

It should be stressed, that although the results of the present meta-analysis support that initiation of anticoagulation with DOACs within the first week of symptom onset does not compromise patient safety, the generalizability of our findings is limited to patients with mild stroke severity and small infarct volumes. Moreover, since most studies included in this meta-analysis *a priori* excluded patients at high risk of haemorrhagic transformation, no inferences regarding potential selection criteria for early anticoagulation can be drawn by the present data. Importantly, most of the data are extracted from observational studies where the use and timing of anticoagulation may have been confounded by both measurable and unmeasurable factors (e.g., physician’s decision making, confounding by indication), further emphasising the need for adequately powered RCTs. Ongoing RCTs investigating the optimal timing for commencing anticoagulant treatment in AF-related IS [ELAN (Early Versus Late Initiation of Direct Oral Anticoagulants in Post-Ischaemic Stroke Patients With Atrial Fibrillation), NCT03148457 [28]; TIMING (Timing of Oral Anticoagulant Therapy in Acute Ischemic Stroke With Atrial Fibrillation), NCT02961348 [29]; OPTIMAS (Optimal Timing of Anticoagulation After Acute Ischaemic Stroke), NCT03759938 [30]; and START (Optimal Delay Time to Initiate Anticoagulation After Ischemic Stroke in Atrial Fibrillation), NCT03021928 [31]] will provide robust data regarding the optimal timing of anticoagulation as well as the preferred anticoagulation strategy in AIS patients with AF.

Certain methodological shortcomings of the present meta-analysis need to be acknowledged. First, limitations conferred by the design of observational studies may have introduced significant selection bias that cannot be accounted for using a meta-analytical approach. Second, the two included RCTs assessed the safety and efficacy of apixaban and rivaroxaban [10,11]; to the best of our knowledge there is no randomized data comparing edoxaban or dabigatran with VKAs, while edoxaban was also under-represented in observational studies. Third, the majority of included studies in this meta-analysis have not explicitly focused on AIS but also included patients with TIA. Fourth, only the study by Seiffge et al. [23] that included individual patient data from seven prospective cohort studies comprised a large study population (4912 patients), whereas the remainder of studies that were meta-analyzed herein had significantly smaller sample sizes. Fifth, our results should be interpreted cautiously due to the noted overlap between time-windows for initiating anticoagulant treatment (i.e., within 1 versus within 2 weeks from AIS), which may have confounded or introduced type II errors to our results. We expect that with the emergence of robust data from the aforementioned RCTs these limitations will be mitigated, allowing the development of comprehensive guidelines for individualized early anticoagulation for patients with AF-related AIS. To this end, it should be stressed that as additional thrombotic, proatherogenic, and proinflammatory factors may confer an increased AIS risk in NOAC-treated patients [32,33], regular reassessment for concomitant factors that may insufficiently respond to NOACs is crucial in the context of individualized AIS prevention strategies.

In conclusion, the findings of the present meta-analysis indicate that early initiation of anticoagulation within the first week from symptom onset in patients with AF-related AIS with mild stroke severity and small infarct size may have a similar efficacy and safety profile compared to later anticoagulation (within 2 weeks). Within the first week after index-AIS, DOACs seem more effective in terms of IS recurrence and survival compared to VKAs, without compromising patient safety. These preliminary observations require independent validation by ongoing RCTs.

## Figures and Tables

**Figure 1 jcm-11-04981-f001:**
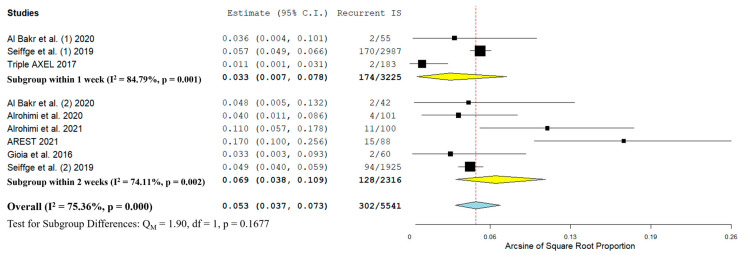
Forest plot presenting the pooled proportion of patients with recurrent IS following the initiation of oral anticoagulants, stratified by the timing of treatment initiation [10,11,18,19,20,21,23].

**Figure 2 jcm-11-04981-f002:**
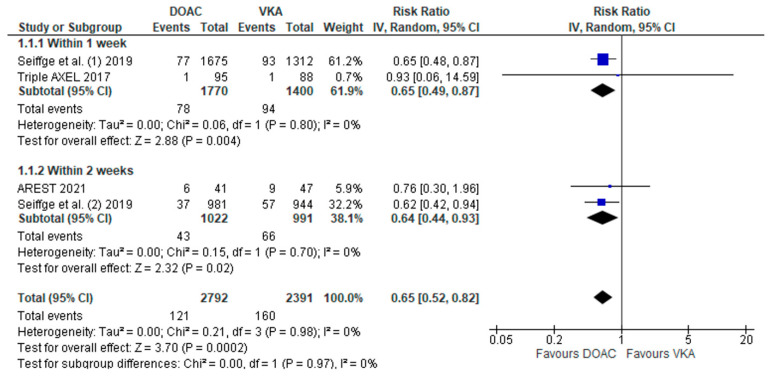
Forest plot presenting the risk ratio of recurrent IS among patients treated with DOACs versus VKAs, stratified by the timing of treatment initiation [10,11,23].

**Table 1 jcm-11-04981-t001:** Characteristics of the included studies.

Study Name	Year of Publication	Country	Study Design	Time Window	Oral Anticoagulant	Number of Patients
Al Bakr et al. [18]	2020	Saudi Arabia	Cohort	within 1 week	any	97
				within 2 weeks		
Alrohimi et al. [19]	2020	Canada	Cohort	within 2 weeks	dabigatran	101
Alrohimi et al. [20]	2021	Canada	Cohort	within 2 weeks	apixaban	100
AREST [11]	2021	US	RCT	within 2 weeks	apixaban versus VKA	88
Gioia et al. [21]	2016	Canada	Cohort	within 2 weeks	any	60
SATES [22]	2020	Italy	Cohort	within 1 week	edoxaban	75
Seiffge et al. [23]	2019	multicenter	IPDM from cohort studies	within 1 week	DOAC versus VKA	4912
				within 2 weeks		
Triple AXEL [10]	2017	South Korea	RCT	within 1 week	rivaroxaban versus VKA	183

RCT: randomized-controlled clinical trials; VKA: vitamin-K antagonists; DOAC: direct oral anticoagulants.

**Table 2 jcm-11-04981-t002:** Overview of the primary outcomes of the single-arm analysis according to the different time windows of initiating oral anticoagulants.

Variable	Time Windows	Prevalence			Test for Subgroup Differences
		*N* of Studies	Pooled Estimates (95%CI)	I^2^, *p* for Cochran Q	
Recurrent IS	Within 1 week	3	3.3% (0.7–7.8%)	85%, 0.001	*p* = 0.1677
	Within 2 weeks	6	6.9% (3.8–6.6%)	74%, 0.002	
	Overall	7	5.3% (3.7–7.3%)	75%, <0.001	
Symptomatic ICH	Within 1 week	4	1.3% (0.3–3.1%)	65%, 0.034	*p* = 0.8941
	Within 2 weeks	6	1.4% (0.7–2.6%)	30%, 0.209	
	Overall	8	1.3% (0.8–2.1%)	49%, 0.039	
Any ICH	Within 1 week	3	27.9% (22.3–33.8%)	19%, 0.293	*p* = 0.0085
	Within 2 weeks	5	11.5% (4.6–21%)	85%, <0.001	
	Overall	7	17% (9–26.9%)	90%, <0.001	
All-cause Mortality	Within 1 week	3	4.6% (0.6–11.9%)	86%, <0.001	*p* = 0.7786
	Within 2 weeks	6	4.3% (1.0–9.8%)	90%, <0.001	
	Overall	7	4.9% (2.8–7.5%)	88%, <0.001	

IS: ischemic stroke; ICH: intracerebral hemorrhage; CI: confidence interval.

**Table 3 jcm-11-04981-t003:** Overview of the primary outcomes of the two-arm analysis comparing DOAC versus VKA in patients on early anticoagulation, stratified by the treatment initiation.

Variable	Time Windows	Effect			Test for Subgroup Differences
		*N* of studies	Risk Ratio (95%CI)	I^2^, *p* for Cochran Q	
Recurrent IS	Within 1 week	2	0.65 (0.49–0.87)	0%, 0.80	*p* = 0.97
	Within 2 weeks	2	0.64 (0.44–0.93)	0%, 0.70	
	Overall	3	0.65 (0.52–0.82)	0%, 0.98	
Symptomatic ICH	Within 1 week	1	0.31 (0.15–0.62)	NA	*p* = 0.53
	Within 2 weeks	2	0.42 (0.21–0.84)	0%, 0.95	
	Overall	2	0.36 (0.22–0.59)	0%, 0.82	
Any ICH	Within 1 week	1	1.1 (0.70–1.71)	NA	*p* = 0.82
	Within 2 weeks	1	0.96 (0.31–2.90)	NA	
	Overall	2	1.08 (0.71–1.63)	0%, 0.82	
All-cause Mortality	Within 1 week	1	0.30 (0.23–0.39)	NA	*p* = 0.003
	Within 2 weeks	2	0.52 (0.40–0.66)	0%, 0.90	
	Overall	2	0.40 (0.25–0.65)	78%, 0.01	

IS: ischemic stroke; ICH: intracerebral hemorrhage; CI: confidence interval.

## Data Availability

The data that support the findings of this study are available from the corresponding author upon reasonable request.

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
