# Peer review of "Early Anticoagulation in Patients with Acute Ischemic Stroke Due to Atrial Fibrillation: A Systematic Review and Meta-Analysis"

_jcm, 2022, doi:10.3390/jcm11174981_

Round 1
Reviewer 1 Report
The authors of this study aimed at investigating the optimal timing for initiation of oral anticoagulation in patients with acute ischemic stroke (AIS) due to atrial fibrillation (AF). To this end, they performed a systematic review and meta-analysis of randomized-controlled trials (RCTs) and prospective observational studies to assess the efficacy and safety of early anticoagulation in AF-related AIS (within 1 week versus 2 weeks). Also, they compared the efficacy and safety of direct oral anticoagulants (DOACs) versus vitamin-K antagonists (VKAs) in the two early time windows. Overall, 8 eligible studies (6 observational, 2 RCTs) were identified, including 5,616 57 patients with AF-related AIS who received early anticoagulation. Patients that received anticoagulants within the first week after index stroke had similar rate of recurrent events compared to patients that received anticoagulation within two weeks. Comparison between DOACs and VKAs showed a significant reduction of stroke recurrences in DOAC-treated patients compared to VKAs. On the basis of these findings, the authors conclude that early initiation of anticoagulation within the first week may have a similar efficacy and safety profile compared to later anticoagulation (within two weeks), while DOACs seem more effective in terms of IS recurrence and survival compared to VKAs.
This is a very interesting meta-analysis that addresses an issue that has been poorly evaluated so far. The manuscript is well-written and easy to ready. English language is fine and figures are self-explanatory.
The authors might want to improve the quality of this investigation by performing additional statistical analysis.
First, they should perform a sensitivity analysis (i.e. repeat all the analysis after excluding each study) in order to verify if results are affected mainly by one investigation.
Second, they should compare results obtained in RCTs with those obtained in observational studies
Finally, they should acknowledge in the Limitation section that only the observational investigation by Seiffge et al included a very large study population.
Reviewer 2 Report
The authors this review performed a systematic analysis of randomized-controlled trials (RCTs) and prospective observational studies to assess the efficacy and safety of early anticoagulation in AF-related AIS (within 1 week versus 2 weeks). A second comparison was performed assessing the efficacy and safety of direct oral anticoagulants (DOACs) versus vitamin-K antagonists (VKAs) in the two early time windows. The outcomes of interest were IS recurrence, all-cause mortality, symptomatic intracerebral hemorrhage (sICH) and any ICH. The authors eight eligible studies (6 observational, 2 RCTs) were identified, including 5,616 patients with AF-related AIS who received early anticoagulation. Patients that received anticoagulants within the first week after index stroke had similar rate of recurrent IS, sICH and all-cause mortality compared to patients that received anticoagulation within two weeks (test for subgroup differences p=0.1677; p=0.8941; and p=0.7786 accordingly). When DOACs were compared to VKAs, there was a significant decline of IS recurrence in DOAC-treated patients compared to VKAs (RR: 0.65; 95%CI: 0.52-0.82), which was evident in both time windows of treatment initiation. DOACs were also associated with lower likelihood of sICH and all-cause mortality.
This review is well written and brings new insight into atrial fibrillation in my opinion.
Nevertheless, in the introduction or discussion section, there is no reference to new studies on the pharmacotherapy of atrial fibrillation.
For example, studies conducted by Wańkowicz et al and Choi et al, emphasize that not only adequate anticoagulation is important, but also a spectrum of completely different risk factors that do not respond to anticoagulation. In my opinion, the inclusion of these studies can further strengthen this review.
Round 2
Reviewer 1 Report
The authors have revised in.-depth the manuscript that was originally submitted.
No further actions are required.
Reviewer 2 Report
I accept this current version of the manuscript